# muSSP: Efficient Min-cost Flow Algorithm for Multi-object Tracking

**Congchao Wang, Yizhi Wang, Yinxue Wang, Chiung-Ting Wu, Guoqiang Yu***
Department of Electrical and Computer Engineering, Virginia Tech
`{ccwang, yzwang, yxwang90, ctwu, yug}@vt.edu`

## Abstract

Min-cost flow has been a widely used paradigm for solving data association problems in multi-object tracking (MOT). However, most existing methods of solving min-cost flow problems in MOT are either direct adoption or slight modifications of generic min-cost flow algorithms, yielding sub-optimal computation efficiency and holding the applications back from larger scale of problems. In this paper, by exploiting the special structures and properties of the graphs formulated in MOT problems, we develop an efficient min-cost flow algorithm, namely, minimum-update Successive Shortest Path (muSSP). muSSP is proved to provide exact optimal solution and we demonstrated its efficiency through 40 experiments on five MOT datasets with various object detection results and a number of graph designs. muSSP is always the most efficient in each experiment compared to the three peer solvers, improving the efficiency by 5 to 337 folds relative to the best competing algorithm and averagely 109 to 4089 folds to each of the three peer methods.

## 1 Introduction

Multi-object tracking (MOT) is a fundamental task in computer vision and has a wide range of applications from traffic surveillance, self-driving cars to cell/particle tracking in microscopy images [7, 20, 15]. In recent years, the min-cost flow formulation of MOT has enjoyed popularity and served as a workhorse in addressing MOT problems [21, 16, 4, 14, 13]. On the one hand, it is a result of the substantial improvement of object detectors which enables the tracking-by-detection strategy [20, 13]. On the other hand, this formulation has great flexibility, for example, it can automatically determine the number of trajectories and deal with missing or spurious detections [15, 4]. Since the min-cost flow problems have been well studied and there exist polynomial-time algorithms [1], it was natural to directly apply the existing algorithms or modify them slightly. Indeed, Zhang et al. [21] used the cost-scaling approach and Pirsiavash et al. [16] proposed to adopt the successive shortest path (SSP) approach. These approaches can guarantee global optimality and are widely considered as the most efficient solvers for generic min-cost flow problems. However, their efficiency is suboptimal by a large margin for the MOT problems [14] as confirmed in our experiments.

With the ever increasing number of objects and duration of tracking, more efficient algorithms for solving MOT min-cost flow problems are in urgent need. In this paper, we identify several important special structures and properties of the graph in the MOT min-cost flow problem and show that they can be used to design efficient algorithms, resulting in a dramatic reduction of computation time. Specifically, the graph in a min-cost flow problem for MOT has the following four specialties: (1) It is a directed acyclic graph (DAG) with single source node $s$ and single sink node $t$. (2) All arcs' capacities are one. (3) Each detection is represented with a pair of nodes, a pre-node and a post-node, with a transition arc between them to incorporate detection confidence. (4) For each detection, the pre-node is linked from $s$ through an inward arc and the post-node is linked to $t$ through an outward arc. Such arcs allow every detection to be the start or the end of a trajectory. An example of a typical min-cost flow graph generated from MOT problem is shown in Fig. 1(b).

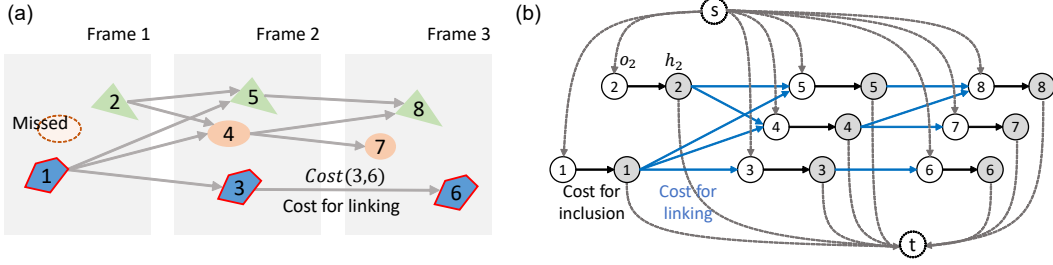

Figure 1: (a) Objects detected in three frames. The first frame has two detections and misses one. Lines between detections are possible ways of linking them. Each line is associated with a cost. Detections 1, 3 and 6 should be linked together as a single trajectory. (b) Typical min-cost flow model for MOT problem. Detection $i$ is represented by a pair of nodes: a pre-node $o_i$ and a post-node $h_i$. The source node $s$ is linked to all pre-nodes and all post-nodes are linked to the sink node $t$. These edges are shown in dashed lines. Edges between detections are shown in blue.

Taking advantage of the specialties, we designed the minimum-update successive shortest path (muSSP) algorithm, an efficient solver for the min-cost flow in MOT problem. While inspired by SSP algorithm [1, 16], muSSP made fundamental changes to its framework by leveraging the observation that, in SSP, most updates in building the shortest path tree through Dijkstra's algorithm were wasted (actually only the shortest $s$-$t$ paths were useful for finding the min-cost flow). As detailed in the method section and Fig. 2 and 3, muSSP consists of four strategies which were guided by theoretical analysis of the graph properties. muSSP follows the philosophy of minimally updating the shortest path tree, only when it is necessary to identify the shortest $s$-$t$ path.

More specifically, firstly, if the cost of linking two detections is too high, we can always have a better association where they are in different trajectories, so these arcs can be safely deleted (**specialty 3 and 4**). Secondly, it can be proved from **specialty 1 and 2** that once an arc connected to either $s$ or $t$ becomes non-empty, the flow in it will always be 1. Pruning these arcs decreases the graph size and, more importantly, makes the following search of $s$-$t$ shortest paths much more efficient. Thirdly, **specialty 1 and 4** enable us to simultaneously augment multiple shortest $s$-$t$ paths. Lastly, since we only have one source and all edges have unit capacity (**specialty 1 and 2**), the nodes to be reviewed for finding shortest $s$-$t$ path can be quickly targeted and form a special tree structure. Dijkstra's algorithm can be modified to leverage this structure, which dramatically decreases the computation.

It appears that muSSP belongs to the solvers of a dynamic single-source shortest path (dSSSP) problem [17, 10, 18]. dSSSP aims to efficiently retrieve the shortest path of each reachable node from the single source node, after the original graph being modified. dSSSP was directly applied to MOT by one of our peer methods, FollowMe [14]. However, our problem is different from the generic dSSSP problem because dSSSP tries to re-build the whole shortest path tree, while we care only the shortest $s$-$t$ path. A large amount of computation is wasted on updating those unrelated vertices.

muSSP has the same theoretical worst-case complexity as SSP, but brings dramatic efficiency boost in real applications. The effectiveness of muSSP is evidenced by forty experiments on five MOT datasets combined with three widely used graph design methods [16, 14, 19]. Because min-cost flow was also frequently used as a sub-routine to approximate the quadratic programming formulation for MOT problems [13, 4], our experiments include both these two kinds of scenarios. Compared with three peer algorithms, SSP, FollowMe [14], and a well-known implementation of cost-scaling algorithm cs2 [12], muSSP is always the most efficient in all experiments, with an efficiency improvement ranging from 5 to 337 folds relative to the best competing algorithm. Regarding to each individual algorithm, muSSP is averagely 4089 times faster than SSP, 1590 times faster than FollowMe, and 109 times faster than cs2. These improvements are achieved without sacrificing the space efficiency.

## 2 Problem Formulation

The problem of associating detections from all frames (data association) can be formulated as a unit capacity min-cost flow problem on a DAG [21, 2] as in Fig.1. In this paper, the term "object" represents a physical object existing over time (e.g. a person), and "detection" indicates a detected snapshot of an object at some time point. An object corresponds to a series (trajectory) of detections.

We denote the graph built in min-cost flow formulation of MOT as $G(V, E, C)$, with node set $V$, arc set $E$, and real-valued arc cost function $C$. The graph has one source $s$ and one sink $t$. For each detection $i$, we create a pre-node $o_i$ and a post-node $h_i$, and three arcs, $(o_i, h_i)$, $(s, o_i)$ and $(h_i, t)$. Any possible spatiotemporal transition of an object is corresponding to some pair of detections, $i$ and $j$ ($i$ before $j$ in time), and an arc $(h_i, o_j)$ is created for it. The capacity of any arc is 1. Any directed path between node $u$ and node $v$ on graph $G$ is denoted as $\pi_G(u, v)$. Under such construction, we can interpret each $s$-$t$ path $\pi_G(s, t)$ as an object trajectory candidate, linking a sequence of detections.

The problem is then turned into selecting a set of object trajectories from all the candidates. In the min-cost flow formulation, this is done by sending flow from $s$ to $t$, and the $s$-$t$ paths eventually with flow inside are selected. Due to unit capacities and the total unimodularity of the problem, the capacity constraints can be reduced to $f_{uv} \in \{0, 1\}, \forall (u, v) \in E$. The set of arc flows is denoted by $f = \{f_{uv} | (u, v) \in E\}$. We ask for in-out balance at any $v \in V \setminus \{s, t\}$ (conservation constraints), so a flow with total integer amount $K$ can only be sent through $K$ distinctive $s$-$t$ paths, reflecting the assumption of non-overlapping object trajectories.

The selection of $s$-$t$ paths is guided by the design of arc cost $C$. MOT looks for object trajectories with stronger evidences of detection, smaller change of a single object across time, and stronger evidences of initial and terminate points. Accordingly, the arc cost $C(o_i, h_i)$ reflects the reward of including the detection $i$. $C(h_i, o_j)$ encodes the similarity between detection $i$ and $j$. $C(s, o_i)$ and $C(h_i, t)$ represent respectively how likely the detection $i$ is the initial point or the terminate point of a trajectory. The flow $f$ with the minimum overall cost $C_{flow}(f) = \sum_{(u,v) \in E} C(u, v) f_{uv}$ is optimal and denoted by $f^*$ (with $f^*_{uv}$ in arc $(u, v) \in E$). The costs can be negative, and therefore min-cost flow formulation automatically leads to the optimal amount of flow when minimizing $C_{flow}(f)$.

This min-cost flow problem can also be formulated in an integer linear programming form:

$$\min_f \sum_{(u,v) \in E} C(u, v) f_{uv} \tag{1}$$

$$\text{s.t.} \quad f_{uv} \in \{0, 1\}, \text{ for all } (u, v) \in E \tag{2}$$

$$\text{and} \quad \sum_{v:(v,u) \in E} f_{vu} = \sum_{v:(u,v) \in E} f_{uv}, \text{ for all } u \in V \setminus \{s, t\}, \tag{3}$$

which is guaranteed to have global optimal solution [1]. In the rest of this paper, the term "graph" and symbol $G(V, E, C)$ all represent the graph built in min-cost flow formulation of MOT problem, and all discussions are specifically for this family of graphs. Besides, the terms node and vertex, edge and arc, shortest path and least-cost path will be used interchangeably.

## 3 Method

muSSP uses four major strategies that take advantage of the special properties of our problem (Fig. 2). We will detail each strategy after giving an overview of the framework. Proofs for lemmas/theorems as well as the time/space complexity analysis can be found in the supplementary.

### 3.1 Overall framework

We first define residual graphs used by min-cost flow solvers before we give an overview of muSSP.
**Definition 1.** The residual graph $G_r(V, E_r, C_r)$ of $G(V, E, C)$ with respect to a flow $f$ is generated by replacing each arc $(u, v) \in E$ by two residual arcs $(u, v) \in E_r$ and $(v, u) \in E_r$, where $(u, v) \in E_r$ has cost $C_r(u, v) = C(u, v)$ and residual capacity $r_{uv} = 1 - f_{uv}$, while $(v, u) \in E_r$ has cost $C_r(v, u) = -C(u, v)$ and $r_{vu} = f_{uv}$.

Given an input graph $G(V, E, C)$, muSSP first removes unnecessary edges in the graph cleaning module. Then an initial shortest path tree ($T_{\mathcal{SP}}$) is obtained. If the stopping criteria is not met, muSSP sends a unit flow from $s$ to $t$ along the shortest $s$-$t$ path in the graph (**AugmentFlow**). The stopping criteria is defined in the same ways as the SSP algorithm in [16]. The multi-path finding module tries to find extra $s$-$t$ paths on which we can send flow without updating the residual graph and shortest path distances (**FindMultiPath**). If it succeeds, we can directly obtain the next $s$-$t$ shortest path (**ExtractPath**). Otherwise, the residual graph is updated based on the $s$-$t$ path found above (**ResidualGraph**). Edges in residual graph that are proved to be never involved in future shortest $s$-$t$ paths are clipped (**ClipPermanentEdge**). The algorithm then updates the shortest path tree and a new $s$-$t$ path is extracted.

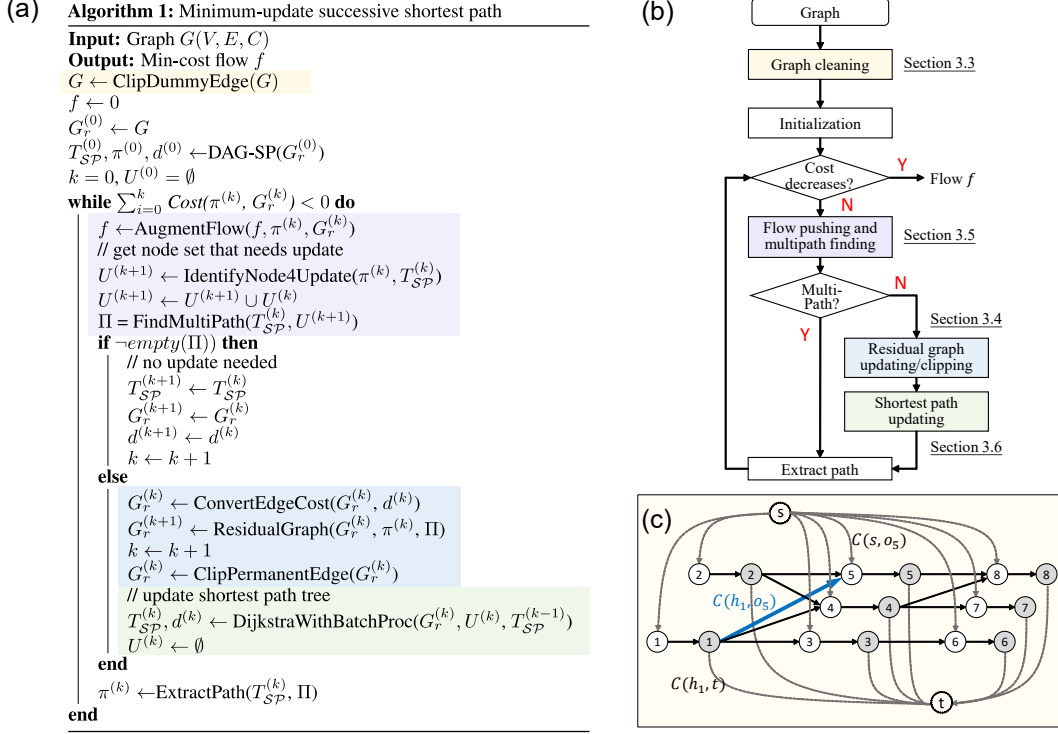

**(a)**

**Algorithm 1:** Minimum-update successive shortest path

**Input:** Graph $G(V, E, C)$
**Output:** Min-cost flow $f$
$G \leftarrow$ ClipDummyEdge($G$)
$f \leftarrow 0$
$G_r^{(0)} \leftarrow G$
$T_{\mathcal{SP}}^{(0)}, \pi^{(0)}, d^{(0)} \leftarrow$ DAG-SP($G_r^{(0)}$)
$k = 0, U^{(0)} = \emptyset$
**while** $\sum_{i=0}^{k} Cost(\pi^{(k)}, G_r^{(k)}) < 0$ **do**
    $f \leftarrow$ AugmentFlow($f, \pi^{(k)}, G_r^{(k)}$)
    // get node set that needs update
    $U^{(k+1)} \leftarrow$ IdentifyNode4Update($\pi^{(k)}, T_{\mathcal{SP}}^{(k)}$)
    $U^{(k+1)} \leftarrow U^{(k+1)} \cup U^{(k)}$
    $\Pi =$ FindMultiPath($T_{\mathcal{SP}}^{(k)}, U^{(k+1)}$)
    **if** $\neg empty(\Pi)$) **then**
        // no update needed
        $T_{\mathcal{SP}}^{(k+1)} \leftarrow T_{\mathcal{SP}}^{(k)}$
        $G_r^{(k+1)} \leftarrow G_r^{(k)}$
        $d^{(k+1)} \leftarrow d^{(k)}$
        $k \leftarrow k + 1$
    **else**
        $G_r^{(k)} \leftarrow$ ConvertEdgeCost($G_r^{(k)}, d^{(k)}$)
        $G_r^{(k+1)} \leftarrow$ ResidualGraph($G_r^{(k)}, \pi^{(k)}, \Pi$)
        $k \leftarrow k + 1$
        $G_r^{(k)} \leftarrow$ ClipPermanentEdge($G_r^{(k)}$)
        // update shortest path tree
        $T_{\mathcal{SP}}^{(k)}, d^{(k)} \leftarrow$ DijkstraWithBatchProc($G_r^{(k)}, U^{(k)}, T_{\mathcal{SP}}^{(k-1)}$)
        $U^{(k)} \leftarrow \emptyset$
    **end**
    $\pi^{(k)} \leftarrow$ ExtractPath($T_{\mathcal{SP}}^{(k)}, \Pi$)
**end**

Figure 2: (a) muSSP algorithm. The four modules/strategies are shown with different shaded colors. (b) Flowchart of muSSP. Each module has the same color as *(a)*. (c) Graph cleaning module for dummy edge removal. The blue arc between $h_1$ and $o_5$ are thicker as it has larger arc cost. If $C(h_1, o_5) > C(s, o_5) + C(h_1, t)$, arc $(h_1, o_5)$ will be removed.

In the initialization step, we use topological sorting to find shortest path (**DAG-SP**). As the graph contains negative-cost edges, we convert the edge cost to reduced cost as $C^d(u, v) = C(u, v) + d(u) - d(v)$ (**ConvertEdgeCost**), where $d(v)$ is the cost of the shortest path from source node $s$ to $v$. The shortest paths found with the reduced cost can be proved to be the same as before [1]. The new edge costs are non-negative and Dijkstra's algorithm can be applied.

## 3.2 The independent flipping lemma

We present a lemma that plays an important role in the muSSP algorithm. Since each time we push a unit flow in a unit-capacity residual graph, the residual graph can be updated by flipping the direction of arcs on the path though which we push flow. The costs of those flipped arcs have the same absolute values as the original ones but have the opposite signs. We denote the set of all simple paths in $G$ from $u$ to $v$ as $\Pi_G(u, v)$. $\Pi_G^*(u, v)$ is the set of all simple least-cost paths in $G$ from $u$ to $v$. The corresponding least cost is denoted as $d_u^*(v)$. Tree is a special type of graph. If tree $T$ is rooted at $v_0$, $\pi_T(v_0, v_i)$ is uniquely determined and we replace it with $\pi_T(v_i)$. A tree $T(V', E', C)$ is embedded in graph $G(V, E, C)$ if $V' \subseteq V$, $E' \subseteq E$.

**Definition 2.** A single-source shortest path tree ($\mathcal{SP}$) is a tree $T(V', E', C)$ embedded in a graph $G(V, E, C)$, rooted at node $s$, such that $\pi_T(v) \in \Pi_G^*(s, v)$ for all $v$ reachable from $s$ in $G$.

Examples of $\mathcal{SP}$ are shown in Fig. 3(a,c,e). Searching for the shortest $s$-$t$ path can be done by maintaining an $\mathcal{SP}$ and dynamically updating it when changes in the graph happen. To update the $\mathcal{SP}$ in a changed graph, we (1) identify the nodes whose lowest-cost paths in the $\mathcal{SP}$ no longer exist in the new graph; (2) update their least-cost paths from $s$ in the new graph and rebuild $\mathcal{SP}$.

**Definition 3.** Given a tree $T(V', E', C)$ rooted at $v_0$, we define the branch node and the set of descendant nodes for a node $v \in V'$: branch($v$) = $u \in V'$ such that $(v_0, u) \in \pi_T(v)$; descendants($v$) = $\{u \in V' | \exists v_x \in V', (v, v_x) \in \pi_T(u)\}$.

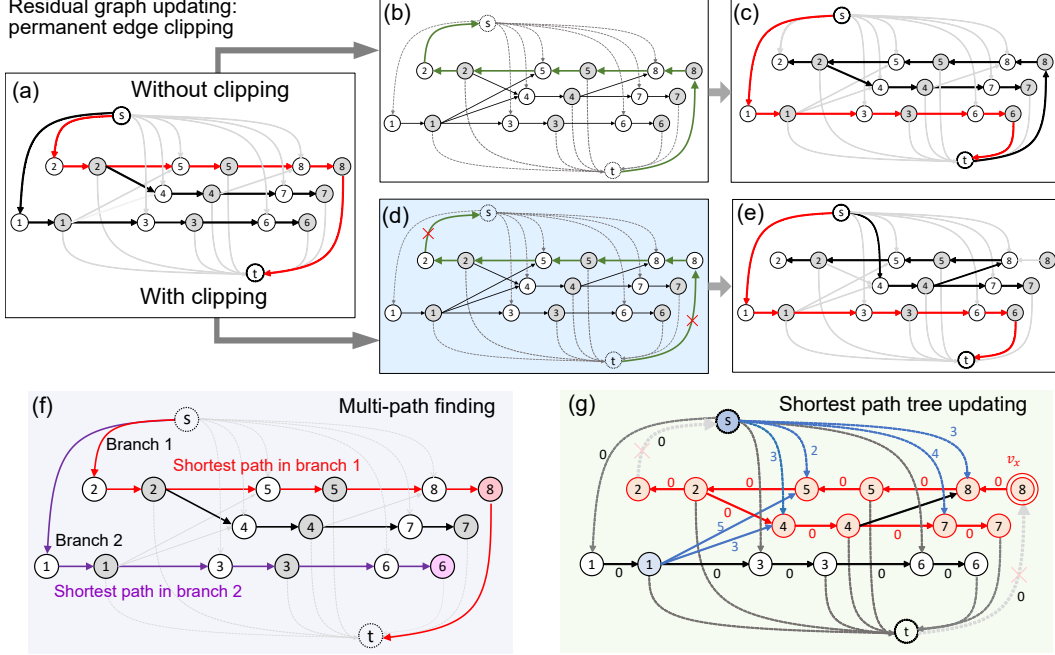

Figure 3: (a) A single source shortest path tree ($\mathcal{SP}$) shown with bold edges. The red path is the shortest $s$-$t$ path. (b) The shortest path in *(a)* is flipped (shown in green). (c) $\mathcal{SP}$ based on the updated residual graph in *(b)*. The red path is the new shortest path. The path through $t$ makes the branch containing the shortest $s$-$t$ path large and we have more nodes to update. (d) Updated residual graph with permanent edges clipped (red crosses). (e) With clipping, the branch containing the shortest path is much smaller than *(c)*. (f) Two branches in the $\mathcal{SP}$, each containing a shortest $s$-$t$ path (red and purple). Only the red one connects to $t$ in $\mathcal{SP}$ as it has a lower cost. We can push flows on both paths without updating residual graph (Theorem 3). Post-nodes $h_8$ and $h_6$ are independent while $h_8$ and $h_7$ are not (Def. 4). (g) After removing permanent edges in the residual graph *(b)*, we need to update the distances from $s$ to the red nodes, because the shortest paths from $s$ to these nodes no longer exist due to the flipping operations. These nodes form a 0-tree. Note that edges in $\mathcal{SP}$ of *(a)* all have zero costs. We initialize the distances from $s$ to them using external edges (in blue) connecting to them. For example, pre-node $o_5$ is linked by arc $(h_1, o_5)$ and $(s, o_5)$, so the initial distance is 2. Based on the initial distance, only a subset of red nodes need to be pushed to the heap (Lemma 7) in Dijkstra's algorithm. Once a node is popped from the heap, we can batch update its descendants. For example, if $o_5$ is firstly popped, all its descendants will be assigned the same distance as $o_5$ (Theorem 4).

Two branches are shown in Fig. 3(f). For a node $v$ in an $\mathcal{SP}$, its branch node is the second node in path $\pi_T(v)$. Here we show that once the shortest $s$-$t$ path is flipped and the residual graph is updated, all nodes whose distances need to be updated share the same branch node.

**Definition 4.** In an $\mathcal{SP}$, two nodes $u$ and $v$ are independent, if and only if branch($u$) $\neq$ branch($v$).

**Lemma 1** (Independent flipping). Given a residual graph $G_r(V_r, E_r, C_r)$ and its $\mathcal{SP}$ denoted as $T$ rooted at $s$, if we flip all edges in path $\pi_T(t)$ and get new graph $G'_r$, for node $v \in V_r$ that is independent with node $t$ in $T$, its least-cost path $\pi_T(v)$ is still valid in the new graph, i.e., $\pi_T(v) \in \Pi^*_{G'_r}(s, v)$.

Lemma 1 shows that when we flip the shortest path $\pi = \{(s, v_0), (v_0, v_1), \ldots, (v_x, t)\}$, only the nodes whose branch node is $v_0$ need update. Reducing the size of descendants($v_0$) directly helps to improve the efficiency. The first two edge clipping strategies we proposed are to fulfill this task.

## 3.3 Graph cleaning with dummy edge clipping

Given a min-cost flow graph $G(V, E, C)$ for MOT problem, an edge $(u, v) \in E$ connecting a post-node to a pre-node is defined as a dummy edge if $C(u, v) > C(s, v) + C(u, t)$ (Fig. 2(c)), In graph cleaning module, this kind of edges are removed to reduce the graph size and the computational cost (**ClipDummyEdge**). Linkage between dissimilar objects tends to be a dummy edge. Therefore, we hypothesize this linkage will not be included in a path, which is proved in the lemma below:

**Lemma 2.** No dummy edges will appear in any optimal solution.

From lemma 2, we can immediately get the following theorem.

**Theorem 1.** Given a graph $G(V, E, C)$ for MOT, removing all its dummy edges does not influence the optimality of the final solution.

### 3.4  Residual graph with permanent edge clipping

After the step of updating the residual graph in each iteration, nodes tend to quickly accumulate to the same branch especially when the graph is sparse (Fig. 3(a,b,c)). The main reason is that after flipping paths, the least-cost path from $s$ to some nodes goes through the sink node $t$. However, we will prove that paths from $t$ to other nodes will never be part of a valid $s$-$t$ shortest path. Therefore, all paths that go through $t$ will be replaced by other paths in future iterations. Thus, the updates of those nodes should not consider the arc originating from $t$. Here we theoretically prove our observation and propose a way to diminish the effects of node accumulation in branches (Fig. 3(d,e)). This is the key step (**ClipPermanentEdge**) in our residual graph updating module.

**Definition 5.** Given a residual graph $G_r(V_r, E_r, C_r)$ in MOT, we define the set of permanent edges as $\{(u, v) \in E_r | v = s \text{ OR } u = t\}$.

Now we show the permanent edges are not necessary in finding a shortest path.

**Lemma 3.** Any $s$-$t$ path with permanent edge is not a simple path.

**Lemma 4.** Given a residual graph, we can always find a shortest $s$-$t$ path which is also a simple path, unless $t$ cannot be reached from $s$.

The above two lemmas imply that we can always find a shortest $s$-$t$ path with no permanent edges.

**Theorem 2.** Given a residual graph $G_r(V_r, E_r, C_r)$, removing all its permanent edges does not influence the optimality of the final solution.

When the flow amount is 0, we have no permanent edge. Each time we instantiate a shortest $s$-$t$ path and flip the arcs on it, we create two permanent edges. From the above theorem, these two edges can be safely removed from current residual graph (Fig. 3(d)). Other than $s$ and $t$, the nodes related to permanent edges can also be removed from the graph, but its contribution to the efficiency improvement is quite limited compared with edge clipping.

### 3.5  Multi-path finding and flipping

Denote the sequentially instantiated shortest paths as $\{\pi_1, \pi_2, \ldots, \pi_K\}$. We observe that the $\mathcal{SP}$ in the $i^{th}$ iteration may contain not only $\pi_i$, but also the following shortest paths $\pi_{i+1}, \pi_{i+2}, \ldots$, with only $t$ missed in them (Fig.3(f)). Simultaneously instantiating and flipping all these shortest paths will save the computation on converting edge costs and decrease the number of duplicated updates, because some nodes may be updated several times if we instantiate the shortest paths one by one (Fig. 3(f)).

Here we analogize the idea of A* algorithm and propose an efficient way to check if current $\mathcal{SP}$ can instantiate more than one shortest path. For each node $v$ in graph $G(V, E, C)$, we define a distance function $d_t(v)$, whose value is $C(v, t)$ if $(v, t) \in E$ and $\infty$ otherwise. Though $d_t$ is not an admissible search heuristic as commonly used in A* algorithm, we can still find the shortest $s$-$t$ path based on it:

**Lemma 5.** Given a residual graph $G_r(V_r, E_r, C_r)$, its $\mathcal{SP}$ rooted at $s$, and the shortest distance function $d$ from $s$ to each node, we have $d(t) = \min(d(v) + d_t(v)), v \in V_r \setminus \{t\}$.

**Theorem 3.** Given a residual graph $G_r(V_r, E_r, C_r)$, its $\mathcal{SP}$ denoted as $T(V'_r, E'_r, C_r)$ rooted at $s$ and a sorted list of $\{d(v) + d_t(v)\}$ with $v \in V_r$ with ascending order, if the $k$ nodes $\{v_1, v_2, \ldots, v_k\}$ that occupy the top $k$ locations of the list are mutually independent, the $k$ paths $\{\pi_T(v_1), \pi_T(v_2), \ldots, \pi_T(v_k)\}$ can be simultaneously instantiated as $k$ shortest paths.

Based on theorem 3, we can efficiently check whether we can instantiate multiple paths from current $\mathcal{SP}$. This is achieved by function **FindMultiPath** given the information of the branches to be updated (**IdentifyNode4Update**). An example of multi-path flipping is shown in Fig. 3(f).

### 3.6 Batch updating and heap shrinking for shortest path tree

In the shortest path tree updating module, we will discuss two approaches that reduce the running time of Dijkstra's algorithm (**DijkstraWithBatchProc**). The first is to simultaneously update the distances of multiple nodes. The second is to push less nodes into the heap used in Dijkstra's algorithm.

If a tree has zero cost for all edges, we call it a 0-tree (Fig. 3(g)). A 0-tree can emerge after edge cost conversion and flipping. We can utilize 0-tree to reduce the computational cost of updating distances of nodes in each iteration. After identifying the nodes that must be updated for finding the next shortest $s$-$t$ path, we can fulfill this task using Dijkstra's algorithm, as the edge costs have been converted to non-negative values. Given $G$ and its $\mathcal{SP}$ denoted as $T$ rooted at $s$, after conversion, $T$ becomes not only an $\mathcal{SP}$ but also a 0-tree. This property can help to accelerate Dijkstra's algorithm. For clarity and simplicity, we assume the updating happens with only one path flipped as is shown in Fig.3(d). The condition of updating nodes after simultaneously flipping multiple paths is the same.

Suppose shortest $s$-$t$ path to be flipped in current iteration is $\pi = \{(s, v_0), (v_0, v_1), \ldots, (v_x, t)\}$. Since the $\mathcal{SP}$ is a 0-tree now, after flipping and permanent edge clipping, the nodes to update can be divided into two sets: $\{t\}$ and another 0-tree rooted at $v_x$ (Fig. 3(g)). As $t$ can be efficiently updated by the sorted list discussed in Theorem 3, here we only show how to update nodes in the 0-tree rooted at $v_x$.

**Lemma 6.** Given a 0-tree $T_0(V_0, E_0, C_r)$ embedded in residual graph $G_r(V_r, E_r, C_r)$, if $v \in$ descendants$(v_0)$, $d_u^*(v) \leq d_u^*(v_0), \forall u \in V_r, \forall v_0 \in V_0$.

Lemma 6 shows that for a node in the 0-tree, the shortest distance from $s$ to its descendants nodes is smaller than or equal to the distance from $s$ to itself. Besides, Dijkstra's algorithm always first deals with node with shorter distances from $s$. Combining these two facts leads to our batch updating strategy below.

**Theorem 4.** In Dijkstra's algorithm, if the distance from $s$ to a node $v$ in a 0-tree is permanently labeled as $d(v)$, $d(v)$ is also the permanent label for the nodes in descendants$(v)$ that haven't been permanently labeled.

It shows that we can permanently label a batch of nodes each time in Dijkstra's algorithm (Fig. 3(g)).

Dijkstra's algorithm is commonly implemented with heap. Decreasing the heap size saves the time consumption of popping/pushing operation on it and increases efficiency. Inspired by the idea in [17] for dSSSP, we found the heap size can also be shrunk based on Lemma 6.

The way we update the 0-tree rooted at $v_x$ with Dijkstra's algorithm is to relax all the edges that link nodes outside the 0-tree to nodes inside it. Then push their temporary distances to the heap. Thus the initial size of the heap will be the node number of the tree. We divide the nodes in the 0-tree into two sets: $P$ and $Q$. Set $P$ contains nodes that cannot be correctly updated without checking the other nodes in the 0-tree and set $Q$ is the complementary set of $P$. Only nodes in $Q$ need to be inserted to the heap. We propose an efficient way to estimate the set of $P$ and only need to insert the remaining nodes in the 0-tree into the heap.

**Lemma 7.** In a 0-tree, nodes with larger temporary distance labels than their parent belong to set $P$.

From Lemma 7, we can use a breadth-first search starting from root $(v_x)$ of the 0-tree, pushing only nodes whose temporary distance labels are not larger than their parents in the 0-tree (Fig. 3(g)). In the best case, the heap size could be 0 and updating of the 0-tree takes linear-time.

## 4 Experiments

We conducted three sets of experiments for the detailed analysis of efficiency improvement. First, muSSP is used to directly solve min-cost flow based data association problems in MOT. Second, we test the efficiency of muSSP as a sub-routine to iteratively approximate the quadratic programming formulation in MOT. Third, to further understand the improvement, we examine the relative contributions of each key strategy we proposed, which can be found in the supplementary.

We compared muSSP with three popular methods, SSP, FollowMe [14], and cs2 [12]. Written in C language, cs2 is an efficient implementation of cost-scaling algorithm, which is widely considered as the best solver for generic min-cost flow problems and was used in [21, 16] for MOT problems.

Table 1: Efficiency comparison on MOT datasets (in seconds)

| Datasets | KITTI(DPM) | | | | KITTI(reglets) | | | |
|---|---|---|---|---|---|---|---|---|
| | seq00 | seq10 | seq11 | seq14 | seq00 | seq10 | seq11 | seq14 |
| (a) graph design from [16] | | | | | | | | |
| SSP | 18.4(108) | 142.8(269) | 68.0(200) | 120.8(448) | 6.2(89) | 9.1(129) | 9.4(156) | 31.8(245) |
| FollowMe | 3.4(20) | 12.7(24) | 6.4(19) | 12.0(44) | 1.4(20) | 1.8(26) | 1.5(25) | 6.4(49) |
| cs2 | 11.8(70) | 88.4(167) | 42.6(125) | 41.1(152) | 4.3(61) | 7.2(103) | 5.7(95) | 10.3(79) |
| **muSSP** | **0.17(1)** | **0.53(1)** | **0.34(1)** | **0.27(1)** | **0.07(1)** | **0.07(1)** | **0.06(1)** | **0.13(1)** |
| (b) graph design from [14] | | | | | | | | |
| SSP | 20.9(116) | 266.9(523) | 77.9(223) | 96.1(291) | 9.4(117) | 12.8(160) | 18.1(227) | 54.7(421) |
| FollowMe | 3.2(18) | 34.4(67) | 9.6(28) | 12.0(36) | 3.8(48) | 3.1(39) | 5.9(74) | 15.3(118) |
| cs2 | 13.6(76) | 98.6(193) | 45.1(129) | 39.8(121) | 5.0(63) | 6.0(75) | 6.3(78) | 8.9(69) |
| **muSSP** | **0.18(1)** | **0.51(1)** | **0.35(1)** | **0.33(1)** | **0.08(1)** | **0.08(1)** | **0.08(1)** | **0.13(1)** |
| (c) graph design from [19] | | | | | | | | |
| SSP | 1.2h(24.5k) | 10.0h(46.2k) | 4.5h(26.0k) | 5.2h(37.6k) | 437.6(4.4k) | 235.8(1.6k) | 246.8(1.8k) | 605.9(3.0k) |
| FollowMe | 917.9(5.1k) | 3.7h(16.9k) | 1.0h(5.5k) | 0.9h(6.7k) | 198.4(2.0k) | 224.5(1.5k) | 242.2(1.7k) | 649.1(3.2k) |
| cs2 | 24.6(137) | 184.3(236) | 78.7(127) | 69.7(139) | 5.9(59) | 8.5(57) | 7.7(55) | 12.8(64) |
| **muSSP** | **0.18(1)** | **0.78(1)** | **0.62(1)** | **0.50(1)** | **0.10(1)** | **0.15(1)** | **0.14(1)** | **0.20(1)** |

| Datasets | CVPR19 | | | | ETHZ(DPM) | | PTC | |
|---|---|---|---|---|---|---|---|---|
| | seq04 | seq06 | seq07 | seq08 | seq03 | seq04 | High | Mid |
| (a) graph design from [16] | | | | | | | (d) probability principled | |
| SSP | 1.8h(4.9k) | 370.9(976) | 23.5(235) | 136.2(619) | 85.7(204) | 173.0(455) | 0.4h(106) | 379.2(134) |
| FollowMe | 2.4h(6.5k) | 801.1(2.1k) | 43.4(434) | 220.6(1.0k) | 21.2(50) | 26.2(69) | 1.2h(339) | 0.3h(430) |
| cs2 | 441.3(337) | 73.7(194) | 16.2(162) | 34.5(157) | 39.4(94) | 46.1(121) | 71.5(5) | 20.2(7) |
| **muSSP** | **1.31(1)** | **0.38(1)** | **0.10(1)** | **0.22(1)** | **0.42(1)** | **0.38(1)** | **13.10(1)** | **2.82(1)** |
| (b) graph design from [14] | | | | | | | Low | |
| SSP | 1.6h(4.3k) | 513.7(1.3k) | 21.2(236) | 38.5(167) | 183.9(400) | 137.3(490) | 10.7(41) | |
| FollowMe | 2.3h(6.2k) | 803.3(2.0k) | 36.2(402) | 62.1(270) | 53.1(115) | 36.8(131) | 40.4(155) | |
| cs2 | 137.2(103) | 35.7(87) | 6.7(74) | 16.0(70) | 63.1(137) | 57.8(206) | 3.0(12) | |
| **muSSP** | **1.33(1)** | **0.41(1)** | **0.09(1)** | **0.23(1)** | **0.46(1)** | **0.28(1)** | **0.25(1)** | |

SSP and our muSSP were implemented in C++. To perform a fair comparison, FollowMe was re-implemented from their python package in C++. The implementation details can be found in the supplementary.

## 4.1 Solving the direct min-cost flow model of data association problems in MOT

To represent the wide range of real world applications, we selected four public datasets including three natural image MOT datasets (ETHZ (BAHNHOF and JELMOLI) [8], KITTI-Car [11], MOT CVPR 2019 Challenge[6]) and one particle tracking dataset (ISBI12 Particle Tracking Challenge (PTC) [5]). For natural image MOT, we designed the graphs using three methods [16, 14, 19]. The difference among them is how to measure the similarity between detections and thus leads to different arc cost functions in the graphs. As [19] is specifically designed for road scene, it was only applied to KITTI-Car dataset. For particle tracking, we used a probability principled way to design the graph, as detailed in supplementary.

All the experiments were based on the detection results either included in the datasets or provided by the authors of [16, 14, 19]. The number of the detections varies from ∼7k to ∼200k. We performed experiments with 39 combinations of detection results, graph design methods, and datasets. Summary of the detected objects, frames and the vertices and edges of the resultant graphs can be found in the supplementary.

Table 1 records the computation time for all experiments on a single core of 2.40GHz Xeon(R) CPU E5-2630. Each cell of the table represents the time consumed by a specific method under a graph design. The time was reported in seconds and sometimes in hours as denoted by the appending letter 'h'. The numbers in the brackets indicate how many folds the method is slower than the most efficient one. Bold font indicates the most efficient method. Overall, muSSP is always the fastest method in all 39 experiments. For majority of the experiments, muSSP achieved sub-second performance, while peer methods need up to several hours. Averagely muSSP is 4193 times faster than SSP, 1630 times faster than FollowMe and 111 times faster than cs2. It can be seen from the table that SSP and FollowMe vary greatly with different graph settings from seconds to several hours. Interestingly, as an improved version of SSP, FollowMe usually performs better than SSP but when the object number is large (e.g., CVPR19), its performance drops quickly and is even worse than SSP. This is because FollowMe does not explicitly know that whether the paths containing flipped edge(s) in current residual graph are all valid in updating the shortest path tree (Lemma 1). To make sure not to miss those valid ones, FollowMe inserts the nodes incident to the flipped edges to the heap used in Dijkstra's algorithm first. This operation will make the heap size huge when we have a large number

of paths flipped. Our experiment on heap shrinking strategy in supplementary clearly shows this. cs2 performs relatively stable and outperforms SSP and FollowMe when the number of objects is large. It is likely that the local updating strategy of cs2 scales better with graph size than SSP and FollowMe. Note that the tracking results comparison is not listed since the solutions obtained by our proposed approach and baseline approaches are the same. This is also true for the following quadratic programming problem where the initialization and step function in Frank-Wolfe algorithm are the same for each method.

## 4.2 Solving the min-cost flow approximation to the high-order modeling in MOT

High-order relationships between detected objects have been incorporated for more accurate tracking, which, however, leads to NP-hard problems [4, 13, 3]. Existing methods approximate the solution with the help of Frank-Wolfe algorithm or Lagrangian relaxation, where min-cost flow solvers were frequently used as a sub-routine. Here we test the efficiency of our muSSP in approximating the solution of a quadratic programming problem formulated in [4] for the tracking problem. The quadratic objective function is provided by the authors in their software package derived from the sequence 'Time_13-59/View_002' of dataset PETS09 S1.L1 [9]. We firstly relax its integer solution constraint and then use Frank-Wolfe algorithm to iteratively solve this quadratic programming problem. In each iteration, the problem is reduced to a min-cost flow problem. The graph has 5866 vertices and 36688 edges, and the algorithm runs totally 400 iterations. Results are shown in Table 2. muSSP is 19 times faster than SSP, 27 times faster than FollowMe and 30 times faster than cs2.

Table 2: Efficiency comparison of solving quadratic programming problem

| Method | SSP | FollowMe | cs2 | **muSSP** |
|---|---|---|---|---|
| PETS S1.L1-2 | 391.9(19) | 557.0(27) | 613.1(30) | **20.3(1)** |

## 5  Conclusion

In this paper, we proposed an efficient yet exact min-cost flow solver muSSP for the data association in MOT problems, taking advantage of the specialties of the graphs built in MOT problems. muSSP is applicable not only to the direct min-cost flow modeling but also to the min-cost flow approximation to the high-order modeling in MOT. The efficiency was demonstrated on a wide range of public datasets combined with various object detection results and graph designs. We expect this large degree of efficiency improvement will save computational time for existing applications, enable engineers to tackle larger scale of problems, and inspire researchers to build more accurate modeling for tracking, for instance, refining iteratively the graph designs based on the tracking results.

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
