[Supplementary Material]

# Supplementary Material for
# muSSP: Efficient Minimum-cost Flow Algorithm for Multi-Object Tracking

## 1   Proofs of theorems and lemmas in the methods section

Any definition, lemma, or theorem is referred to using the same index in the main body and in this supplementary material, if it is discussed in both of them. The major goal of the proofs is to show that with the fundamental changes to the graph or to SSP framework, the optimality of final solution is not influenced.

### 1.1   Independent flipping lemma

**Definition 1.1a.** The cost associated with a directed path $\pi$ is

$$\text{pathcost}(\pi) = \begin{cases} \sum_{(u,v)\in\pi} C(u,v) & \text{if } \pi \neq \emptyset \\ \infty & \text{otherwise.} \end{cases}$$

**Lemma 1** (Independent flipping)**.** Given a residual graph $G_r(V_r, E_r, C_r)$ and its $\mathcal{SP}$ denoted as $T$ rooted at $s$, if we flip all edges in path $\pi_T(t)$ and get new graph $G'_r$, for node $v \in V_r$ that is independent with node $t$ in $T$, its least-cost path $\pi_T(v)$ is still valid in the new graph, i.e., $\pi_T(v) \in \Pi^*_{G'_r}(s,v)$.

*Proof.* Assume there is a new shortest path $\pi_{G'_r}(s,v)$ from $s$ to node $v$ after flipping, which is shorter than $\pi_T(v)$. It is obvious that this path contains flipped edge(s). We can cut $\pi_{G'_r}(v)$ into 4 parts $\{(s,\ldots,v_1),(v_1,\ldots,v_2),(v_2,\ldots,v_3),(v_3,\ldots,v)\}$. Part 1 and 3 contain only edges existing in $E_r$ and part 2 contains only flipped edges. Except the 4th part, the other three can not be empty. Because $(v_1,\ldots,v_2)$ are flipped edges in $T$, we have $d_T(v_2) + pathcost(v_2,\ldots,v_1) \leq pathcost(s,\ldots,v_1)$. For the same reason, $d_T(v_3) \leq d_T(v_2) + pathcost(v_2,\ldots,v_3)$. After adding these two inequalities together, we have $d_T(v_3) \leq pathcost(s,\ldots,v_1) - pathcost(v_2,\ldots,v_1) + pathcost(v_2,\ldots,v_3)$, which is $d_T(v_3) \leq d_{G'_r}(v_3)$. If the 4th part is empty, the proof is done. If not, we can further cut $(v_2,\ldots,v)$ into 4 parts and continue until the 4th part is empty. Then we can have $d_T(v) \leq d_{G'_r}(v)$, which means $\pi_T(v)$ is still the shortest path in $G'_r$. $\square$

### 1.2   Dummy edge clipping

**Lemma 2.** No dummy edge will appear in any optimal solution.

*Proof.* Assume an optimal solution $f^*$ contains a dummy edge $(u,v)$, which is in $s$-$t$ path $\pi = \{(s,\ldots,u),(u,v),(v,\ldots,t)\}$. It is clear that $f_{sv} = f_{ut} = 0$. From the definition of dummy edge, we know that by separating $\pi$ into two paths $\{(s,\ldots,u),(u,t)\}$ and $\{(s,v),(v,\ldots,t)\}$, the total cost of $f^*$ can be further decreased. This contradicts the assumption. $\square$

**Theorem 1.** Given a graph $G(V, E, C)$ for MOT, removing all its dummy edges does not influence the optimality of the final solution.

*Proof.* For a min-cost flow problem on a graph $G$, assume $G'$ is the same graph as $G$ but clipping all dummy edges. It is clear that the optimal solution $f^*_{G'}$ on $G'$ is a feasible solution on $G$. For an optimal solution $f^*_G$ on $G$, because it does not contain dummy edges, it is also a feasible solution to $G'$. Thus, the optimal solutions on $G$ and $G'$ are equivalent. $\qquad\square$

## 1.3 Permanent edge clipping

**Lemma 3.** Any $s$-$t$ path with permanent edge is not a simple path.

*Proof.* An $s$-$t$ path should start at $s$ and end at $t$. If the path contains an edge that leaves $t$, it needs to eventually come back to $t$, which creates a cycle. The same reasoning applies to the case when the path contains an edge that go toward $s$. $\qquad\square$

**Lemma 4.** Given a residual graph, we can always find a shortest $s$-$t$ path which is also a simple path, unless $t$ cannot be reached from $s$.

*Proof.* Our original graph is DAG with no cycles. Because we only augment flow along the shortest path in the residual graph, this will never generate negative-cost cycle in residual graph [1]. Removing cycles in a path will not increase its cost, so for any non-simple $s$-$t$ path, we can find a simple one at least as good as it. $\qquad\square$

**Theorem 2.** Given a residual graph $G_r(V_r, E_r, C_r)$, removing all its permanent edges does not influence the optimality of the final solution.

*Proof.* Assume this residual graph $G_r$ is generated in iteration $i$ and we totally need $K$ iterations. We first prove that the $i$th iteration to $K$th iteration in SSP, it is equivalent to solve a new min-cost flow problem with excess flow number $K - i + 1$ on $G_r$. This is obvious from the paradigm of SSP algorithm. Now we prove that for the new min-cost flow problem, removing permanent edges does not influence the optimality of the final solution. We denote the graph without permanent edges as $G_{r \backslash p}$. From lemma 3 and 4, we know that we can have at least one optimal solution $f^*$ to the new min-cost flow problem on $G_r$ with only simple paths. Assume the optimal solution on $G_{r \backslash p}$ is $f^o$ if there is one. It is clear that $f^o$ is a feasible solution to $G_r$, and we have $C_r * f^o \geq C_r * f^*$. Because $f^*$ contains no permanent edges, $f^*$ is also a feasible solution for $G_{r \backslash p}$ and $C_r * f^o \leq C_r * f^*$. Then we get $C_r * f^o = C_r * f^*$, so the optimality are not influenced. $\qquad\square$

## 1.4 Multi-path finding

**Lemma 5.** Given a residual graph $G_r(V_r, E_r, C_r)$, its $\mathcal{SP}$ rooted at $s$, and the shortest distance function $d$ from $s$ to each node, we have $d(t) = \min(d(v) + d_t(v)), v \in V_r \setminus \{t\}$.

*Proof.* For any $s$-$t$ path $\pi = \{(s, \dots, v_x), (v_x, t)\}$, pathcost$(\pi) \geq d(v_x) + d_t(v_x)$. Thus we have $d(t) \geq \min(d(v) + d_t(v)), v \in V_r$. Based on the definition of shortest path and $d_t(v)$, we have $d(t) \leq d(v) + d_t(v)$, for any $v \in V_r$. Thus $d(t) = \min(d(v) + d_t(v)), v \in V_r \setminus \{t\}$. $\qquad\square$

By maintaining a sorted list $\{d(v) + d_t(v)\}$ in ascending order, we can directly extract the shortest $s$-$t$ path by popping the top element in the list. The updating of sink $t$ after path flipping can also be fulfilled by updating this list.

**Lemma 1.4a.** Given a graph $G(V, E, C)$ with source node $s$ and sink node $t$, and its $\mathcal{SP}$ $T$ rooted at $s$ with shortest distance function $d$, flipping the shortest $s$-$t$ path will never decrease distance $d(v)$ in the new graph, for any $v \in V$.

*Proof.* The proof is similar to lemma 1. From lemma 1, we know this is true for nodes that independent with node $t$. Here we only consider nodes that inside the same branch with $t$. Assume the graph with flipped path is $G_r$. There is a new shortest path $\pi_{G_r}(v)$ in $G_r$, from $s$ to node $v$, whose cost is smaller than $\pi_T(v)$. It is obvious that this path contains flipped edge(s). We can cut $\pi_{G_r}(v)$

into 3 parts $\{(s, \ldots, v_1), (v_1, \ldots, v_2), (v_2, \ldots, v)\}$. Part 1 contains only edges existing in $E$ and part 2 contains only flipped edges. Part 1 can not be empty. If part 2 and 3 are empty, the lemma is correct. If part 2 is not empty, from $\mathcal{SP}$ $T$, we have $d_T(v_2) + pathcost(v_2, \ldots, v_1) \leq pathcost(s, \ldots, v_1)$, which means $d_T(v_2) \leq d_{G_r}(v_2)$. If part 3 is empty, $v_2 = v$, the proof is done. If part 3 is not empty but contains no flipped edges, we have $d_T(v_2) + pathcost(v_2, \ldots, v) \geq d_{T(v)}$. After adding the previous inequality together, we have $d_T(v) \leq d_{G_r}(v)$. If part 3 is not empty and contains flipped edges, we can continue previous cutting strategy on part 3 until the new part 3 is empty or contains no flipped edges. Then we can have $d_T(v) \leq d_{G_r}(v)$, which means the cost of the new shortest path does not decrease. $\square$

This lemma shows that every time when we find a shortest $s$-$t$ path, flipping it will not create a shorter path in the residual graph. Before flipping the shortest path, the node occupying the top of the sorted list $\{d(v) + d_t(v)\}$ denotes current shortest path that should be instantiated. After flipping, the top element is popped out. After that, if the new top node of the list is independent with the popped one, it will hold that position and become the next shortest path to be flipped. Based on the same reasoning. We can generalize this reasoning in the following theorem.

**Theorem 3.** Given a residual graph $G_r(V_r, E_r, C_r)$, its $\mathcal{SP}$ denoted as $T(V_r', E_r', C_r)$ rooted at $s$ and a sorted list of $\{d(v) + d_t(v)\}$ with $v \in V_r$ with ascending order, if the $k$ nodes $\{v_1, v_2, \ldots, v_k\}$ that occupy the top $k$ locations of the list are mutually independent, the $k$ paths $\{\pi_T(v_1), \pi_T(v_2), \ldots, \pi_T(v_k)\}$ can be simultaneously instantiated as $k$ shortest paths.

*Proof.* From lemma 5, we know $v_1$ indicates current shortest path $\pi_1$. We only need to prove that for any $i \in \{1, \ldots, k\}$, after popping the nodes $\{v_1, \ldots, v_i\}$ in the list, the $i+1$ node will occupy the top of the list. Because the $k$ nodes are mutually independent, $v_{i+1}$ is independent with $\{v_1, \ldots, v_i\}$. From lemma 1, we know $d(v_{i+1})$ does not change in the residual graph after $i$ iterations. From lemma 1.4a, we know $d(u)$ does not decrease for any $u \in V_r$, so after popping out $\{v_1, \ldots, v_i\}$, in the new graph $v_{i+1}$ will occupy the top location of the list and indicate the new shortest path. $\square$

## 1.5 Batch updating and heap shrinking

**Lemma 6.** Given a 0-tree $T_0(V_0, E_0, C_r)$ embedded in residual graph $G_r(V_r, E_r, C_r)$, if $v \in$ descendants($v_0$), $d_u^*(v) \leq d_u^*(v_0), \forall u \in V_r, \forall v_0 \in V_0$.

*Proof.* Because $v \in$ descendants($v_0$), there is a path from $v$ to $v_0$ in $T_0$ with only 0-cost edges. Thus $d_u^*(v) \leq d_u^*(v_0), \forall u \in V_r, \forall v_0 \in V_0$. $\square$

**Theorem 4.** In Dijkstra's algorithm, if the distance from $s$ to a node $v$ in a 0-tree is permanently labeled as $d(v)$, $d(v)$ is also the permanent label for the nodes in descendants($v$) that haven't been permanently labeled.

*Proof.* Based on the property of Dijkstra's algorithm, for a permanently labeled node $v$, its distance $d(v) \leq d(u)$ for any node $u$ with temporary label. If $u \in$ descendants($v$), $d(v) \geq d(u)$ based on lemma 6. Thus, we can permanently label node $u$ as $d(v)$. $\square$

**Lemma 7.** In a 0-tree, nodes with larger temporary distance labels than their parent belong to set $P$.

*Proof.* The temporary distance of a node $v$ is from the relaxation of edges linked from nodes outside of the 0-tree. If it is larger than the temporary distance label of its parent node $u$, based on lemma 6, $v$ will be updated at least once more by relaxing edge $(u, v)$. Thus $v$ cannot be correctly labeled without checking nodes in the 0-tree. $\square$

# 2 Complexity analysis

## 2.1 Time complexity analysis

Min-cost flow in MOT problem can be solved by successive shortest path (SSP) algorithm with computational complexity $O(K(n \log(n) + m))$ [7], where $n$ and $m$ are the number of vertices and arcs, respectively. We now show that the complexity of muSSP algorithm does not increase

compared with SSP. The complexity of graph cleaning and finding the shortest path in DAG is $O(m)$, which only run once. The loop will terminate with $K$ iterations. For one iteration, identifying the nodes to update takes $O(n)$ operation by checking their branch nodes. Finding multi-paths for flipping can be efficiently done by maintaining the list mentioned in section 3.5. The list can be implemented by priority queue with updating complexity $O(n \log(n))$. Converting edge costs takes $O(m)$ time, while building residual graph and clipping permanent edges take amortized constant time. The best complexity for implementing Dijkstra's algorithm is $O(n \log(n) + m)$. Consequently, the complexity in one iteration is still dominated by Dijkstra's algorithm as $O(n \log(n) + m)$ and the overall complexity is $O(K(n \log(n) + m))$. The worst case for muSSP happens when the shortest-path tree contains only one branch in each iteration of Alg. 1. This worst case is unlikely to happen in the graph of MOT problem, as the number of branches is at least equal to the number of objects in the first (or last) frame of the video.

## 2.2    Space complexity analysis

The efficiency improvement is achieved without scarifying memory cost. Given a graph with the number of nodes $n$ and number of edges $m$, the memory consumption for the graph itself is $O(m+n)$. Dummy edge clipping and permanent edge clipping use $O(1)$ space. For each node, we save a branch node label, parent node label, distance label of each node which need $O(n)$ space. The list and heap used in Dijkstra algorithm in multi-path flipping also need $O(n)$ space. Thus our memory consumption is totally $O(m + n)$, which is the same as SSP and FollowMe[6].

# 3    Experiments

## 3.1    Effectiveness of each individual strategy

For different graphs, the contribution of these strategies incorporated in muSSP varies significantly. First, we check the number of nodes updated in each iteration, and Fig.1 shows the comparison of our first three strategies and the other methods on the two sample graphs. Details of the graphs can be found in the captain of the figure. "DC" and "PC" are short for "dummy-edge clipping" and "permanent-edge clipping", and "DC-PC" represents these two strategies are applied simultaneously. We can see that by clipping dummy/permanent edges, the number of nodes to update dramatically decreases compared with FollowMe or SSP. Because SSP can have an early stop once it finds the shortest $s$-$t$ path [1], the updated number of nodes is not necessarily to be the same as node number in the graph. In fig.1a, we did not show the result of "DC-PC", because in this graph, there is no dummy edge. DC will not contribute to decrease the number of nodes for updating. However, in fig.1b, there is a clear difference between curve "PC" and "DC-PC" especially in the early iterations. Purple spikes denote the iterations updating is really conducted. Gaps between purple spikes mean we find those shortest $s$-$t$ paths without updating the shortest path tree. After applying multi-path flipping, the number of updating shortest path tree decreases from 470 to 123 in the graph used in fig.1a and 466 to 106 in the graph used in fig.1b.

Using the same graph as fig.1a, we show the power of batch updating (BU) and heap shrinking (HS) in table 1. Here "muSSP" incorporates the first 3 strategies. "muSSP-BP" includes the batch updating and "muSSP-BU-HS" includes both the batch updating and heap shrinking. The heap we are using are based on binary search trees, which has amortized constant complexity for popping and $\log(n)$ for pushing. Thus, the heap operation here only considers pushing operation. Clearly, purely with the first 3 strategies, the number of nodes for updating has already significantly decreased. With batch updating, the number of heap operations decreases for one more order. The initial heap size can also be decreased to half with heap shrinking. Here we do not consider the number of heap operations in building the initial heap because for SSP, it is always 1. FollowMe indeed suffers a lot from the initial heap size. They insert every node in the previously flipped path to the heap first. This operation will make the heap huge when we have a large number of paths.

## 3.2    Implementation details

Except for cs2, muSSP, SSP, and FollowMe are all implemented in C++. The source code of these 3 algorithms can be found on GitHub(`https://github.com/yu-lab-vt/muSSP`). We use the data structure of self-balanced binary search tree to implement the Dijkstra's algorithm which

Figure 1: The effectiveness of dummy-edge clipping ("DC"), permanent-clipping ("PC") and multi-path flipping on two different graphs. Gaps between purple spikes denote the iterations we skipped because of multi-path flipping. The graph used in (a) is from sequence 07 in MOT CVPR19 dataset with graph design method in [6]. The graph in (b) is from ISBI12 Particle Tracking Challenge (sequence name: "RECEPTOR snr 7 density low") with probability principled graph design described in supplementary.

Table 1: Comparison of # heap operation and initial heap size

| Avg. of all iterations | SSP | FollowMe | muSSP | muSSP-BU | muSSP-BU-HS |
|---|---|---|---|---|---|
| #Heap operation | 44889.5 | 6225.1 | 49.6 | 5.8 | 5.7 |
| Initial heap size | 1.0 | 33273.9 | 85.4 | 85.4 | 39.4 |

168  has $O(1)$ for popping top element and $O(log(n))$ for pushing a new value. This is the reason why
169  we mainly compare pushing operation number used in Dijkstra's algorithm when testing the fourth
170  strategy.

171  For muSSP, SSP and FollowMe, it is not needed to explicitly set flow number. For cs2, we use binary
172  search to find the best flow number as did in [11, 7].

173  Except for cs2, the other 3 methods accept real-valued edge cost and saved using "double" data type.
174  For cs2, we round the edge costs with accuracy to 1e-7. More details about implementation can be
175  found in the attached source code. The cs2 package can be found on the author's website [4].

176  All comparisons were conducted on Ubuntu 16.04 LTS, compiled by g++ v5.4.0 with single core of
177  2.40GHz Xeon(R) CPU E5-2630 and memory speed at 2133MHz.

## 3.3   Detection results used for graph building

179  For KITTI-Car dataset, we chose four long sequences (seq00, seq10, seq11, seq14) for a clear
180  efficiency comparison. For sequences that too small, there is no need to speed up for any solver.
181  KITTI provides two detection results DPM[3] and reglets[10]. We test on both of them. For MOT
182  CVPR 2019 dataset, we use its test set with 4 videos. Each video contains a crowd of pedestrians.
183  The detection results are provided by their website. The detection results of ETHZ (BAHNHOF and
184  JELMOLI) dataset is also DPM from [7]. Particle tracking challenge provided two different types of
185  simulated particle tracking data: RECEPTOR and VESICLE, each of them contains 15 data with
186  different signal to noise ratio and particle density. We test our efficiency on the RECEPTOR snr 7
187  with density from low to high. As this dataset does not provide detection results, we directly use the
188  ground truth as input to build the graph. The data RECEPTOR snr 4 was used as training data.

189  For the quadratic programming problem, we directly use the function formulated in the published
190  software package [2]. We use Frank-Wolfe algorithm to approximate the solution. The step size is set
191  as $k/(k + 2)$, where $k$ is the index of the current iteration. We set a stringent stopping criterion and
192  with max iteration number as 400.

Table 2: Details of KITTI-Car dataset

| Datasets | KITTI(DPM) | | | | KITTI(reglets) | | | |
|---|---|---|---|---|---|---|---|---|
| | seq00 | seq10 | seq11 | seq14 | seq00 | seq10 | seq11 | seq14 |
| #frames | 465 | 1176 | 774 | 850 | 465 | 1176 | 774 | 850 |
| #detections | 51100 | 181132 | 104748 | 96974 | 19885 | 22189 | 24524 | 35198 |
| (a) graph design from [7] | | | | | | | | |
| #vertices | 102202 | 362266 | 209498 | 193950 | 39772 | 44380 | 49050 | 70398 |
| #arcs | 171135 | 608881 | 349869 | 325256 | 71730 | 78273 | 86864 | 123008 |
| (b) graph design from [6] | | | | | | | | |
| #vertices | 102202 | 362266 | 209498 | 193950 | 39772 | 44380 | 49050 | 70398 |
| #arcs | 173242 | 609576 | 352140 | 328352 | 71977 | 78524 | 87048 | 122875 |
| (c) graph design from [8] | | | | | | | | |
| #vertices | 102202 | 362266 | 209498 | 193950 | 39772 | 44380 | 49050 | 70398 |
| #arcs | 172317 | 607035 | 350276 | 326635 | 71592 | 78477 | 86830 | 122763 |

Table 3: Details of CVPR19, EHTZ and PTC datasets

| Datasets | CVPR19 | | | | ETHZ | | PTC | | |
|---|---|---|---|---|---|---|---|---|---|
| | seq04 | seq06 | seq07 | seq08 | seq03 | seq04 | High | Mid | Low |
| #frames | 2080 | 1008 | 585 | 806 | 1000 | 936 | 101 | 101 | 101 |
| #detections | 208000 | 70189 | 20220 | 43444 | 101180 | 94054 | 77352 | 39215 | 7438 |
| (a) graph design from [7] | | | | | | | probability principled | | |
| #vertices | 416002 | 140380 | 40442 | 86890 | 202362 | 188110 | 154706 | 78432 | 14878 |
| #arcs | 1007552 | 334881 | 98255 | 206836 | 358546 | 365461 | 462213 | 234438 | 44448 |
| (b) graph design from [6] | | | | | | | | | |
| #vertices | 416002 | 140380 | 40442 | 86890 | 202362 | 188110 | | | |
| #arcs | 1007436 | 334513 | 98256 | 206557 | 411170 | 368080 | | | |

## 3.4 Graph design

We use three existing methods [7, 6, 8] to build the graph. Details can be found in their paper and published software packages. [7, 6] are purposely designed with the min-cost flow framework. For [8], in their package, they design similarity between detections and use Hungarian algorithm [5] to solve the association problem in every two consecutive frames. The cost they used is just the negative values of the similarities. We use the same way for linking cost between detections. In [8], they believe in detection results and majority of the detections have links in adjacent frames. Following the same thinking, we set the transition edge between the pre-node and the post-node of one detection as zero. The enter/exit costs are set as a small value to avoid trivial solutions. To tackle the problem of occlusion or missed detection, for all these datasets, we allow the objects to have one "jump", which means we allow the objects in frame $k$ to be linked to objects in frames $k + 1$ and $k + 2$.

For PTC dataset, we use the ground truth as input. Because particles can suddenly appear or disappear in the filed of view, we set the cost of $C(s, v) = -\log(p_{enter})$ and $C(u, t) = -\log(p_{exit})$ for any pre-node $v$ and post-node $u$. The probabilities $p_{enter}$ and $p_{exit}$ are learned from another data provided in the challenge (the data "RECEPTOR snr 4"). Because we are using the groundtruth as input, we set the linkage cost between the pre-node $v_i$ and post-node $u_i$ of the same object as $C(v_i, u_i) = -(C(s, v_i) + C(u_i, t))$ to make sure the final results will not miss any detection. The linkage costs between different objects are decided by their distance as we do not have much appearance features to use as in natural image data. We firstly learn an empirical distribution of the real distance distribution from training data (the data "RECEPTOR snr 4"). Thus, for each distance value, we can have its p-value $p$ based on this distribution. Similarly, we set the linkage cost as the $-\log(p)$. As in [9], each detection is linked to its 3 nearest neighbors in the next frame.

## 3.5 Details of the data

The details of the data we experimented on can be found in Table 2 and 3. It includes the numbers of frames and detections in each video. After combined with different graph design methods, the out-coming graph sizes are also listed.