[Reviews · NeurIPS 2019]

Reviewer 1



Summary: The paper presents a new efficient solver for the min-cost flow framework that solves the MOT problem. The efficiency was achieved by exploiting special graph structures in the MOT problem and saving unnecessary computations that a more general solver would perform. The authors showed the efficiency of the new solver by comparing the running time of the new method to that of the existing solvers on multiple tracking sequences. Strengths: -- made good observations regarding where computations can be saved by carefully analyzing the flow formulation. -- provided an analytical insight regarding how the solution optimality is still maintained by the proposed method. Also, time and space complexity analyses were provided in the supplementary document. Weaknesses: -- Only running time comparisons were provided in the experiment section. Thus, it is hard to know how well the method performs in terms of outputting accurate object trajectories. It would be worth reporting numbers on other tracking metrics such as MOTA, IDF1, and IDS as well. Also, are the solutions found by SSP, FollowMe, cs2, and the propose method (muSSP) always the same? If so, please highlight the fact that the final solutions obtained by the proposed method and the baselines were all the same in Table 1. For the results in Table 2, I believe that the solutions are not the same anymore because the formulation used in Table 2 does not allow us to find the global optimal solution. In this case, the comparison for tracking accuracy is required. -- The worst time complexity of the new solver is the same as that of SSP. However, the new solver does not seem to reach the worst case in the experiments that the authors reported. What would be the case that the new solver will struggle to find the optimal solution and thus become slow? -- The paper presentation could be improved such that it would be easier for the readers to follow. Overall comment: The paper did not provide any analysis regarding the tracking accuracy of the proposed approach and the baseline approaches. The authors may have not included this in the paper because there is no difference in terms of tracking accuracy between the proposed method and the baselines. However, the authors could still compare it against other recent state-of-the-art approaches in order to show that the min-cost flow solvers achieve strong tracking performance and that it is thus worth investing on designing more efficient flow solvers for the MOT problem. Without seeing the tracking accuracy results, it is very hard for me to judge if the proposed method is an effective solution for MOT. Final comment: Some of my concerns were addressed by the authors' rebuttal. However, my biggest concern still remains. The authors did not provide any results of the proposed approach in terms of the tracking accuracy in the paper and in the supplemental document. Thus, I cannot know whether or not the tracking results that the authors generated for Table 1 and 2 are reasonably good. Because of this reason, I will keep my original score (4) for this paper. I would like to encourage the authors to provide more detailed analysis including the tracking accuracy in the paper in order to make the paper stronger.

Reviewer 2



This paper tackles a fairly difficult problem, namely data association for multi-object tracking and provides a novel algorithm based on a min-cut, max-flow formulation. Given the complexity of data association for MOT, I found the technical presentation to be especially clear (and insightful). I also give high marks for the scholarship and demonstrated familiary with both the problem and the literature. Rather than blindly applying standard algorithms to the problem, they provide a thorough analysis of the complexity of the graph and how it may be pruned in way that achieves significant computational savings. While one might view this as a very specialized result, I think the detailed analysis would be of interest outisde multi-object tracking for other problems that might benefit from graph simpification. They provide analysis and guarantees related to the various algorithmic changes. They provide convincing empirical comparisons to existing algorithms. It was the case that I had to reread various sections due to the density of the material, but I think that is the nature of this particular problem. Minor questions/comments: 1) if an object is detected in frames i-1 and i+1, but missed in i, can the algortihm recover the trajectory. It seems like the answer would be no since a flow cannot be established. 2) line 261 "We designed the graphs using three different methods" - What does this mean? I assumed the graph was based on a set of detections. Is this just saying that depending on the detection method, you would get a different graph (seems obvious) or something else? Please clarify. Post Rebuttal: The authors addressed the minor questions that I raised. I still feel strongly that this paper merits inclusion. It contributes to both multi-object tracking in a practical and important way *AND* the analysis provides insight into other network flow problems.

Reviewer 3



The paper is clearly written, and the theoretic part is solid. The experiments also show the efficiency of the proposed algorithm. Minor concerns: The running time of different algorithm can be influenced by the CPU brand, compiler options, memory speed, \etc. The authors can give a detailed spec of the platform in the appendix. ==== After rebuttal ==== I think the author resolves the reviewers' concern successfully. I tend to vote accept for the paper.

[Author Response · NeurIPS 2019]

We thank the reviewers for their thoughtful feedback. The major concern was **whether current min-cost flow (MCF)**
**based methods can achieve strong tracking performance and thus worth investing in designing efficient flow**
**solvers**. We would like to take the opportunity to report the evidences and explanations to relieve this concern and
emphasize the significance of our work.

First, **the strong performance of MCF based methods is evidenced** by the tracking results on two public benchmarks,
MOT17 and KITTI-Car. (1) As shown in Table R1, the MCF based tracking methods (LSST17 and JBNOT) are the top
2 methods among the 32 published works on MOT17 benchmark, according to the MOTA score which is widely used
to evaluate tracking performance. It is worthy to mention, among all the 99 submissions including anonymous ones,
LSST17 still holds the first place. (2) On KITTI-Car benchmark (Table R2), MCF based methods also achieve quite
competitive performance, where MOTBeyondPixels and AB3DMOT take the first two places among all 34 published
works according to MOTA score. These comparisons are provided by MOT challenge [R1] and KITTI challenge [R2],
where the corresponding references can be found.

Table R1: Top 5 methods on MOT17*

| | **MOTA** | IDF1 | MT | ML | ID Sw. | MCF |
|---|---|---|---|---|---|---|
| **LSST17** | **54.7** | **62.3** | 20.4 | 40.1 | **1243** | YES |
| JBNOT | 52.6 | 50.8 | 19.7 | 35.8 | 6050 | YES |
| FAMNet | 52.0 | 48.7 | 19.1 | **33.4** | 3072 | NO |
| eTC17 | 51.9 | 58.1 | 23.1 | 35.5 | 2288 | NO |
| eHAF17 | 51.8 | 54.8 | **23.4** | 37.9 | 1834 | NO |

\* Anonymous submissions are excluded as their method details are not accessible

Table R2: Top 5 methods on KITTI-Car*

| | **MOTA** | MOTP | MT | ML | IDS | MCF |
|---|---|---|---|---|---|---|
| MOTBeyondPixels[†] | **84.2** | **85.7** | **73.2** | **2.8** | 468 | YES |
| AB3DMOT | 83.8 | 85.2 | 66.9 | 11.4 | **9** | YES |
| aUToTrack | 82.3 | 80.5 | 72.6 | 3.5 | 1025 | NO |
| JCSTD | 80.6 | 81.8 | 56.8 | 7.4 | 61 | NO |
| 3D-CNN/PMBM | 80.4 | 81.3 | 62.8 | 6.2 | 121 | NO |

[†] This graph design method was used in our submitted paper

Second, **investing in speeding up MCF solver is valuable to the whole field**, based on the following facts. (1)
Computational efficiency has been identified as the bottleneck for MCF-based methods, as pointed out by references [14,
16, R3]. Indeed, [R3] turned to non-MCF based approach because the current MCF solvers could not meet their time
requirement. Researchers in [14,16] developed faster sub-optimal approaches by sacrificing some tracking accuracy,
as they were not satisfied with the global but slower MCF solvers. Our work guarantees global optimality and offers
hundreds to thousands times of improvement over existing solvers. Thus, we believe this work would significantly
improve the efficiency of the existing MCF based methods and enable many otherwise infeasible applications. Reference
14 and 16 are listed in our submitted paper. (2) As pointed out by Reviewer 2, the proposed strategies in muSSP are not
limited to MOT problem. Except "dummy edge clipping", all the other strategies are applicable to generic unit-capacity
min-cost flow problem. muSSP may also serve as an inspiration to develop specialized but more efficient network flow
based algorithms.

**Specific responses to Reviewer 1:**

(a) The tracking results obtained by our proposed approach and baseline approaches are all the same, since MCF
formulation admits a unique global optimal solution and we replace old MCF solvers by our more efficient one. For
the quadratic problem (Table 2 in the paper), it can be proved that with the same initialization and step function in
Frank-Wolfe method, the final solution is the same.

(b) The theoretical worst case of muSSP happens when the shortest-path tree contains only one branch in each iteration
of Alg.1. This worst case is unlikely to happen in the graph of MOT problem, as the number of branches is at least
equal to the number of objects in the first (or last) frame of the video. We will include these clarifications in the paper.

**Specific responses to Reviewer 2:**

(a) The MCF framework is able to handle missing detections as it allows linkages between detections beyond adjacent
frames (e.g. from i-1 frame to i+1 frame).

(b) With the same set of detections, the nodes in the graph are fixed, but the arcs are not. Actually, the flexibility of
setting different arcs and assigning different weights leads to various algorithms for MOT problems. For example,
in [16], the weights of arcs were designed based on detection confidence, while [14, 20] used extra features such as
detections' appearance similarity. We will include these clarifications in the paper.

**Specific responses to Reviewer 3:**

All comparisons were conducted on Ubuntu 16.04 LTS, compiled by g++ v5.4.0 with single core of 2.40GHz Xeon(R)
CPU E5-2630 and memory speed at 2133MHz. This will be added in the revision.

# References

[R1] The MOT17 Tracking Challenge. https://motchallenge.net/results/MOT17/
[R2] The KITTI Tracking Challenge. http://www.cvlibs.net/datasets/kitti/eval_tracking.php
[R3] Yoon, Young-Chul, et al. arXiv preprint arXiv:1907.00831 (2019).


[Meta-Review · NeurIPS 2019]

• The paper presents a method on efficient min-cost flow algorithm for the multi-object tracking problem. The main idea of the min-cost flow algorithm is to remove unnecessary edges in the graph in order to find a more efficient solution in term of time. Then through a series of operation on the shortest path tree, it poposes a good solutions. Initially reviewers disagree in the rate with a very satisfied reviewer ( rating it 9), a very negative one ( rating it 4) and a positive reviewer . This reviewer was satisfied on the rebuttal rating it 7 at the end. One reviewer remains not convinced since in the paper there are no evaluation in term of accuracy that is a strong point for multiple-target tracking. After the discussion the area chair agrees for the acceptance